# The Immune Efficacy of Inactivated Pseudorabies Vaccine Prepared from FJ-2012ΔgE/gI Strain

**DOI:** 10.3390/microorganisms10101880

**Published:** 2022-09-21

**Authors:** Qiu-Yong Chen, Xue-Min Wu, Yong-Liang Che, Ru-Jing Chen, Bo Hou, Chen-Yan Wang, Long-Bai Wang, Lun-Jiang Zhou

**Affiliations:** 1Institute of Animal Husbandry and Veterinary Medicine, FuJian Academy of Agriculture Sciences, Fuzhou 350013, China; 2Fujian Animal Disease Control Technology Development Center, Fuzhou 350013, China

**Keywords:** pseudorabies virus, inactivated vaccine, vaccine efficacy, control

## Abstract

An emerging pseudorabies virus (PRV) variant has been reported on Bartha-K61-vaccinated farms since 2011, causing great economic losses to China’s swine-feeding industry. In this study, two vaccines, FJ-2012ΔgE/gI-GEL02 and FJ-2012ΔgE/gI-206VG, were administered to piglets for immune efficacy investigation. Humoral immunity response, clinical signs, survival rate, tissue viral load, and pathology were assessed in piglets. The results showed that both vaccines were effective against the PRV FJ-2012 challenge, the piglets all survived while developing a high level of gB-specific antibody and neutralizing antibody, the virus load in tissue was alleviated, and no clinical PR signs or pathological lesions were displayed. In the unimmunized challenged group, typical clinical signs of pseudorabies were observed, and the piglets all died at 7 days post-challenge. Compared with commercial vaccines, the Bartha-K61 vaccine group could not provide full protection, which might be due to a lower vaccine dose; the inactivated vaccine vPRV* group piglets survived, displaying mild clinical signs. The asterisk denotes inactivation. These results indicate that FJ-2012ΔgE/gI-GEL02 and FJ-2012ΔgE/gI-206VG were effective and could be promising vaccines to control or eradicate the new PRV epidemic in China.

## 1. Introduction

Pseudorabies virus (PRV) is a member of the *Herpesviridae* family (subfamily *Alphaherpesvirinae*) and is the pathogen that causes pseudorabies (PR) [1]. The main clinical signs of PR for pigs are characterized by three syndromes: central nervous system (CNS) disorders in piglets, abortion in pregnant swine, and respiratory disorder signs in finishing pigs [2,3,4]. Although PRV can infect numerous mammals such as wild boar, cow, goat, dog, sheep, mink, and fox [5,6,7,8,9], the pig is the primary host and the only reservoir of PRV. Discovered in Hungary in 1902, PRV infection has caused huge economic losses worldwide [10].

Before the emergence of variant PRV strains in 2011, Bartha-K61, imported from Hungary in the 1970s, was the first live, gene-modified vaccine widely used in China. Subsequently, two gene-modified vaccines based on the classical PRVs isolated from China were developed: SA215, based on the Fa strain, was licensed in 2003 [11], and HB, based on the Ea strain, was licensed in 2006 [12]. Since then, PR has been largely controlled by the wide use of inactivated and live-attenuated vaccines, and some farms eradicated PR via the concept of DIVA, i.e., the serological differentiation of vaccinated from field-virus infected pigs by the use of marker vaccines and respective companion diagnostic tests.

However, since late 2011, PR outbreaks have been reported on Bartha-K61-vaccinated farms and have spread rapidly. In response to the major challenge of the new PRV variant, a large number of novel genetically engineered vaccines based on emerging PRV variants were developed, including inactivated and live-attenuated vaccines (shown in Table 1). These vaccines were reported to provide effective protection against PRV variant infections, but so far only two vaccines, PRV HeN1201 and PRV C strains, have been licensed. Furthermore, from the PRV live virus-vectored vaccines, which encode key antigens of other animal pathogens such as the VP2 gene of porcine parvovirus type 2 (PCV2), the E2 genes of the classical swine fever virus (CSFV) were created, and these candidate vaccines provided effective protection against multiple infectious diseases, including PR [13,14]. None of the PRV vector virus vaccines have so far attained the status of practical application or been commercialized [15]. Amazingly, the confirmed recombinant expressing gD- or gC/gD-based protein vaccines could work against pseudorabies virus infection, but these candidate vaccines are not yet commercially available [16,17].

Until now, both the licensed vaccines based on classical and variant PRV strains had been used to control PR in China [33], but there was some debate about whether Bartha-K61 could protect pigs against the Chinese variant PRV strains in a past study [34]. On the one hand, some research showed that despite differences in virulence, a suitable vaccination scheme with the Bartha-K61 strain could protect pigs against variant PRV challenge by XJ5, HeN1, or Hercules [35,36]. On the other hand, some studies showed that Bartha-K61 was unable to provide complete protection against challenge by the variants, TJ, ZJ01, SMX, or JS-2012 [20,23,27,37], but the vaccine failure mechanism was not clear. Therefore, in the absence of a new, powerful candidate vaccine specific to these PRV variants, it was urgent that more efficient vaccines be developed.

In our previous report, we isolated the highly virulent PRV strain FJ-2012, a confirmed antigenic variant from classical PRV strains, which caused higher mortality and more serious pathological changes [38]. Then, our laboratory prepared two gE/gI-deleted PRV-inactivated vaccines based on the PRV FJ-2012 strain. In this study, we explored the immune efficacy of the inactivated vaccine in piglets to develop a novel vaccine to control and eradicate the new PRV variants.

## 2. Materials and Methods

### 2.1. Experimental Suckling Pig and Sites

A total of 30 14-day-old Duroc × Landrace × Yorkshire (DLY) suckling pigs from a family pig farm in Fujian province were screened for experimentation in the Animal Laboratory of the Fujian Academy of Agricultural Sciences. Gum swabs and blood samples were collected to confirm that they were CSFV, PRV, and PRRSV pathogen negative by RT-PCR (Appendix A) and PRV gB and gE negative (Appendix A) via serological antibody ELISA before immunization.

### 2.2. Experimental Vaccines and Main Reagents

FJ-2012ΔgE/gI-GEL02 (FJ-2012-GEL*) and FJ-2012ΔgE/gI-206VG (FJ-2012-VG*) inactivated vaccines were prepared, and MONTANIDE^TM^ GEL02 and ISA 206 adjuvants were purchased from SEPPIC (Special Chemicals Co., Ltd., Shanghai, China). Bartha-K61 (K61) trivalent attenuated vaccine, 5000 TCID_50_ a dose, and inactivated vaccine PRV Ea strain (vPRV*) were purchased from Animal Husbandry Industry Co., Ltd. (Chengdou, China). The PRV gB and PRV gE antibody ELISA kit was purchased from IDEXX Laboratories, Inc. (Westbrook, ME, USA). FJ-2012 was isolated from brain samples collected from a Bartha-K61-vaccinated farm in Fujian, southern China, that had a PR outbreak [38]. Viral DNA purified by a QIAamp DNA Mini kit purchased from Qiagen Co., Ltd (Hilden, Germany). TaqMan^TM^ Fast Advanced was purchased from ThermoFisher Scientific (Waltham, MA, USA). TaqMan Real-time PCR testing was conducted using an ABI StepOne Plus real-time PCR instrument. The virus was multiplied on PK-15 cells, cultured in Dulbecco’s modified Eagle’s medium (DMEM, Hyclone, UT, USA) containing 1% fetal bovine serum (FBS, Gibco, CA, USA), 100 IU/mL penicillin, and 100 μg/mL streptomycin (PS, Gibco, CA, USA) at 37 °C and 5% CO_2_. Ten percent formalin was purchased from Sangon Biotech Co., Ltd. (Shanghai, China).

### 2.3. Experimental Pigs Groups and Immunization Procedures

The 30 suckling pigs were randomly divided into six groups of five: four immunization groups, one unimmunized challenged group, and one control group. The schedule and timepoint design of immunization and challenge in different groups are shown in Table 2. The pigs in FJ-2012-GEL* and FJ-2012-VG* groups were intramuscularly (i.m.) immunized with 2 mL inactivated vaccine prepared from FJ-2012ΔgE/gI-GEL102, and FJ-2012ΔgE/gI-206VG at 0 and 21 days. The vPRV* group was intramuscularly (i.m.) injected with 2 mL commercial inactivated vaccine of Ea strain at 0 and 21 days. In group K61, the pigs were intramuscularly (i.m.) immunized with one dose of 5000 TCID_50_ live-attenuated Bartha-K61 vaccine at 0 days. The pigs in the unimmunized challenged group were injected with 2 mL sterile 0.9% NaCl. The control group was left untreated.

### 2.4. Experimental Pigs Challenged Procedures and Clinical Record

At 42 days after the first immunization, all pigs in the FJ-2012-GEL*, FJ-2012-VG*, vPRV*, K61, and unimmunized challenged groups were challenged intranasally (i.n.) with 3 mL 10^6^ TCID_50_ FJ-2012 per piglets. The pigs in the control group were not challenged, serving as a negative control. After being challenged, survival rate and rectal temperatures were measured once daily (morning), and the clinical signs of each piglet were observed and scored twice (morning/evening) daily according to the previous report [39]: clinical scores were assessed as (1) elevated temperature above 40 °C and below 41 °C; (2) fever above 41 °C, combined with respiratory distress; (3) ataxia; (4) convulsions; and (5) moribund or dead. The higher of the two scores each day was assigned as the individual daily score.

### 2.5. PRV gE, gB Antibody Test, and Neutralizing Antibody Detection

Blood samples were collected at 0, 14, 21, 28, 35, 42, 49, and 56 days after the first immunization, and the serum was separated. PRV gE and gB antibodies were detected, by an antibody test kit purchased from IDEXX Laboratories, Inc. (ME, USA), according to the manufacturer’s instructions. Serum samples taken at 21, 35, and 42 days after the first immunization were subjected to a neutralization test, as previously described [20]. For the neutralization antibody assay, dilutions of serum were co-incubated with the virus (0.2–200 TCID_50_) for 1 h at 37 °C prior to the addition of PK cells in DMEM (Hyclone, UT, USA) containing 1% FBS and 1% PS. After 4 days of incubation at 37 °C and 5% CO_2_, the neutralization was analyzed by observing cytopathic effects (CPEs) under a microscope. The highest dilution well with no CPE was noted to be the PD_50_ for the antibody. The neutralizing antibody titer was calculated using the Reed–Muench method.

### 2.6. Necropsy, Tissue Sampling, and Histological Analysis

At 14 days post-challenge (dpc), all surviving pigs were euthanized by injections of tiletamine hydrochloride and zolazepam hydrochloride (Virbac, Carros, France). A complete necropsy was performed within 2 h of death, and tissue samples required for histological examination—lung, kidney, tonsil, brain, and spleen—were obtained from each. These samples were fixed in 10% formalin, processed routinely, and embedded in paraffin. Histological examination was carried out by microscope (Olympus BX53, Japan) and image analysis (SC180, Japan). Microscopic lesions were evaluated in a blinded fashion by two veterinary pathologists (Wang Quanxi and Qi Boming, FAFU). Three visual fields were randomly selected, and a score scale was established following the Chunlian Song [40] and Opriessnig, T. [41] methods. Lung sections were scored for the presence and severity of alveolar septal infiltration with inflammatory cells, the amount of alveolar exudate, and the amount of pulmonary congestion ranging from 0 to 6 (0, normal; 1, mild multifocal; 2, mild diffuse; 3, moderate multifocal; 4, moderate diffuse; 5, severe multifocal; 6, severe diffuse). Kidney sections were evaluated for the presence of lymphohistiocytic inflammation and scored from 0 (none) to 3 (severe). Tonsil and spleen sections were evaluated for the presence of lymphoid depletion ranging from 0 to 3 (0, normal; 1, mild lymphoid depletion with a loss of overall cellularity; 2, moderate lymphoid depletion; 3, severe lymphoid depletion with a loss of lymphoid follicle structure) and the presence of inflammation ranging from 0 to 3 (0, normal; 1, mild histiocytic-to-granulomatous inflammation; 2, moderate histiocytic-to-granulomatous inflammation; 3, severe histiocytic-to-granulomatous inflammation with the replacement of follicles), and the scores of lymphoid depletion and inflammation were summed. Brain sections were scored for the presence of lesions, and the scores were summed. Lesion present (score 1): Inflammatory cell infiltration; neuronal degeneration; glial cell lesion (increased, nodular); eosinophilic inclusion bodies; organizational change; neuronal loss. Lesion absent (score 0).

### 2.7. Establishment of PRV qPCR Method and Tissue Viral Load Analysis

Tissue samples of lung, brain, and tonsil, required for PRV viral load detection, were stored at −80 °C. The viral DNA was purified from a 1 g tissue sample using the QIAamp DNA Mini Kit (Qiagen, Germany) according to the manufacturer’s instructions. Specific primers and probes were designed based on the gE gene of PRV (Genbank: NC_006151.1), gE-F: TGGGCTCCTTGGGCGTATCA, gE-R: GTCTCGCCGRTGCAGTAG, and gE-Probe: FAM-TCTGGCTGTGCGAGTCTGCT CC-BHQ1. The qPCR test was conducted at a total volume of 20.0 μL containing 10.0 μL 2× TaqMan Fast Advanced Master Mix (TaqMan^TM^ Fast Advanced, ThermoFisher, MA, USA), 2.0 μL DNA, 6.5 μL sterilized H_2_O, and 0.5 μL each of the primers (final primer concentration 0.5 mM). The reaction was heated to 95 °C for 1 min, followed by 40 cycles of 95 °C for 5 s, and 60 °C for 20 s (fluorescence captured). TaqMan real-time PCR was carried out using the ABI StepOne Plus real-time PCR instrument. The target DNA segment of the gE gene was cloned into the Pmd18-T vector (TaKaRa, Japan), recombinant plasmid (T-PRV) identified by sequencing at Sangon Biotech Co., Ltd. (Shanghai, China). A 10-fold serial dilution of the T-PRV was performed with different concentrations (10^1^–10^6^ copies/uL) of positive plasmids as templates, and three replicates for each gradient plasmid were established. The standard logarithm (standard curve) was derived using the common logarithm (lgC) of the standard starting copy number as the abscissa and the cycle threshold (Ct value) as the ordinate, which was used for the calculated viral load in tissue samples. According to the CT value of each tissue sample, the original copies were calculated by the standard curve and converted into copies of the virus load in each gram of tissue with multiples (25×). The copy numbers of each tissue sample were expressed as log_10_ copies per gram of tissue sample.

### 2.8. Statistical Analysis

Data were analyzed using SPSS 16.0 and GraphPad Prism version 6.0. All dates were presented as mean ± SD. Pairwise comparisons of neutralizing antibody titers between different groups were performed by one-way ANOVA, followed by Tukey’s Multiple Comparison test; microscopic lesions score and differences in viral DNA load in different tissue samples were measured by one-way repeated measurement analysis of variance (ANOVA) and the least significant difference (LSD). Differences were considered statistically significant if *p* < 0.05.

### 2.9. Ethical Statement

All experimental procedures were conducted in accordance with the regulations of the Administration of Affairs Concerning Experimental Animals, approved by the Laboratory Animal Bioethics Committee of the Institute of Animal Husbandry and Veterinary Medicine, FAAS, in accordance with animal ethics guidelines and protocol. The approval numbers of the ethics committee were IAHV-AEC-2021-066.

## 3. Results

### 3.1. PRV gB and gE Antibody Production in Piglets

After vaccination, the piglets in each group showed normal appetites and displayed no clinical signs. Serum samples taken from all piglets were tested for gB and gE antibodies at 0, 14, 21, 28, 35, 42, 49, and 56 days post-vaccination (dpv). As shown in Figure 1A, before being challenged, PRV gB antibodies in all vaccinated piglets in groups FJ-2012-GEL*, FJ-2012-VG*, vPRV*, and K61, were detected at 28, 21, 21, and 14 days post-vaccination, respectively, while no gB antibodies were detected in the unimmunized challenged or control groups. After the PRV FJ-2012 challenge, all piglets in the unimmunized group tested positive for the gB antibody at 7 dpc. Meanwhile, no gE antibody was detected in any of the vaccination groups before the PRV FJ-2012 challenge. Afterwards, the gE antibody was developed in the vaccinated groups and unimmunized challenged group at 7 dpc (49 dpv). No gB or gE antibodies were observed in the control group (Figure 1B).

### 3.2. Virus Neutralizing Antibodies of PRV Induced by Vaccines in Piglets

The serum samples of piglets in each group were collected at 21, 35, and 42 dpv, and the virus neutralizing antibody (VNA) titer against PRV FJ-2012 after immunization is shown in Figure 2. VNAs were detected at 21 days post first immunization, at which time a treated booster immunization was given. The VNAs increased progressively until 42 dpc. The piglets vaccinated with FJ-2012-VG* developed higher levels of neutralizing antibodies compared with the other vaccination groups at each time point and reached the highest titer among all vaccination groups (*p* < 0.01). The neutralizing antibody titers of the attenuated vaccine group K61 at 21 days were higher than in the treated FJ-2012-GEL* and vPRV* groups (*p* < 0.01), while the neutralizing antibody titers at 35 dpc were lower than FJ-2012-VG* (*p* < 0.05) and vPRV* (*p* < 0.01), as well as at 42 dpc (*p* < 0.01), indicating a rapid neutralization in the K61 group and the inefficient production of neutralizing antibodies.

### 3.3. Protection of Vaccinated Piglets against Virulent Challenge

There were no abnormal rectal temperatures or adverse clinical signs before the challenge. After 42 days post first immunization, all piglets except those in the control group were inoculated intranasally (i.n.) with 3 mL 10^6^ TCID_50_ FJ-2012 strain. All piglets in the FJ-2012-GEL* and FJ-2012-VG* groups had a transient fever 1 to 3 dpc, but returned to normal temperature (Figure 3A) without other clinical signs (Figure 3B), and all survived (Figure 3C). In the vPRV* group, all piglets had a rectal temperature as high as 41 °C from 2 to 3 dpc. Subsequently, the fever reduced slightly but remained above 40 °C until 8 dpc (Figure 3A). Mild clinical signs including depression, anorexia, and sleepiness also appeared. The clinical score was only raised at 2 and 3 dpc (Figure 3B). Ultimately, all piglets survived (Figure 3C).

However, in the K61 group, the piglets displayed a high rectal temperature (above 41 °C) 2 to 4 dpc, and it continued above 40 °C until 11 dpc (Figure 3A). Compared with those in the vPRV* groups, these piglets presented severe clinical signs of depression, anorexia, and respiratory distress. The clinical score of the K61 pigs increased from 1 to 5 dpc (Figure 3B). One of these pigs displayed convulsions, cough, and diarrhea, and then died at 13 dpc. The remaining K61 pigs showed only transient clinical signs and survived (Figure 3C). Meanwhile, the piglets in the unimmunized challenged group after the FJ-2012 challenge displayed typical clinical signs of pseudorabies (PRV): rectal temperature over 41 °C (Figure 3A), respiratory distress, depression, anorexia, and neurological symptoms. The clinical score was higher from 2 dpc to death or euthanasia compared to the other groups (Figure 3B). Two and three of the piglets died on 5 and 7 dpc, respectively (Figure 3C). The piglets in the control group had normal rectal temperatures and clinical signs, and all survived (Figure 3A–C).

### 3.4. Necropsy and Histological Analysis

The surviving piglets were euthanized on day 14 post-challenge. Necropsies were performed on them and the piglets that died during the challenge. The incidence rate of tissue lesions in groups was summarized in Table 3. Hemorrhaging, defuse reddened foci, and edema lesions were observed in the lung, and the incidence rate of the unimmunized challenged group (5/5) is higher than other groups. The lesions of the kidney exhibited petechiation, and only can be observed in the unimmunized challenged group (3/5). The tonsil lesions showed mucosa swelling and bleeding, with a layer of yellow-white exudate on the surface, observed in three piglets in the unimmunized challenged group and two pigs in the K61 group. Cerebral edema and meningeal hyperemia were observed in the piglets of the unimmunized challenged group (5/5) and the K61 group (1/5). The hemorrhagic necrosis lesions in the spleen were observed in the unimmunized challenged group (2/5). In contrast, the tissue of the control, FJ-2012-GEL*, and FJ-2012-VG* piglets displayed no pathological lesions.

To study the pathological condition of these organs further, tissues from the lung, kidney, tonsil, brain, and spleen were stained with hematoxylin and eosin (H.E). Microscopic lesion scores of the tissue are summarized in Table 4. The piglets in the unimmunized challenged group showed multiple lesions in several organs, such as severe pulmonary congestion in the lungs, hemorrhaging, and cellular serous exudates in the bronchiolar cavities, which were red-stained (Figure 4(A5)), and mild infarction and congestion in the kidneys (Figure 4(B5)). The tonsils degenerated and became necrotic (Figure 4(C5)). Non-suppurative ganglioneuritis characterized by intranuclear eosinophilic inclusions, hemorrhage, and pronounced perivascular inflammatory infiltrates were observed in brain tissue; meanwhile, vascular sheath morphology formed (Figure 4(D5)). The number of splenic lymphocytes decreased; red pulp severely congested spleen tissues; and multiple small focal necroses were discovered (Figure 4(E5)). The MLS of the unimmunized challenged group was higher than other groups (*p* < 0.05). However, the histopathological results of these organs from FJ-2012-GEL* and FJ-2012-VG* showed no significant pathological changes (Figure 4(A1–E1,A2–E2)). Among commercial vaccine groups, pulmonary abscesses, hemorrhages, cellular serous exudates (Figure 4(A4)), slight renal congestion (Figure 4(B4)), tonsillar cell modification (Figure 4(C4)), inflammatory brain cell infiltration (Figure 4(D4)), and slightly decreased splenic lymphocytes (Figure 4(E4)) were observed in the piglets of the K61 group. The MLS of the lung, kidney, tonsil, brain, and spleen is significantly different compared with the FJ-2012-GEL* and FJ-2012-VG* groups (*p* < 0.01). The vPRV* group piglets only showed cellular serous exudates in the bronchiolar cavities in the lungs (Figure 4(A3)). The lung MLS for the vPRV* group was 2.00 ± 1.00, and for the FJ-2012-GEL* and FJ-2012-VG* groups was 0.00 ± 0.00 (*p* < 0.05).

### 3.5. Tissue Viral Load Analysis

A standard linear regression equation was derived from the lgC of the standard starting copy number as the abscissa and the Ct value as the ordinate. It was determined to be Y = −3.119X + 38.761, the correlation coefficient R^2^ = 0.998, and the reaction efficiency for measuring the viral load of the tissue samples was 109.2%. As shown in Figure 5, the viral genomic copy numbers in the lungs, kidneys, and tonsils of the unimmunized challenged group were significantly higher than in the vaccinated groups (*p* < 0.01). Their brains and spleen were significantly different from those of the inactivated vaccine groups (FJ-2012-GEL*, FJ-2012-VG*, and vPRV* (*p* < 0.01)), and the activated vaccine group K61 (*p* < 0.05). The viral genomic copy numbers in all tissues of the attenuated vaccine group (K61) were higher than in the inactivated vaccine groups (*p* < 0.05), especially in the brain, tonsils, and spleen (*p* < 0.01). Moreover, among the inactivated vaccine groups, the viral copy numbers in the lung, tonsil, and brain of the vPRV* group were significantly different from FJ-2012-GEL* and FJ-2012-VG* (*p* < 0.05). For the viral copy numbers in lung and tonsil tissues, the vPRV* group was significantly different from FJ-2012-GEL* and FJ-2012-VG* (*p* < 0.01). The copy number of the viral genome in brain tissue was significantly different from FJ-2012-GEL* (*p* < 0.05) and FJ-2012-VG* (*p* < 0.01). However, the viral copy numbers in the kidney and spleen were not significantly different among the inactivated vaccine groups (*p* > 0.05). There were no significant differences in all tissue samples between the FJ-2012-GEL* and FJ-2012-VG* groups (*p* > 0.05).

## 4. Discussion

Since 2011, despite Bartha-K61 vaccination, outbreaks of PR have re-emerged on many pig farms in China leading to a fatality rate of up to 50% in infected newborn piglets and increased virulence in older pigs [37,38,42,43]. Viral genome analysis has shown that PRVs are classified into the following two genotypes according to phylogenetic characteristics: Genotype I contains isolates from Europe and America, while Genotype II contains isolates primarily from China and other Asian countries [44]. Novel variant strains such as HeN1, TJ, and FJ-2012 belong to Genotype II, the genome of which showed large variations—substitutions, insertions, and deletions—compared to Genotype I [38,45,46], which may explain why PR re-emerged, but the detailed mechanism of its incomplete protection remains unclear.

In past studies, there was some debate about whether Bartha-K61 could protect pigs against variant PRV strains [34]. In this study, we evaluated the immune efficiency of FJ-2012-VG* and FJ-2012-GEL* vaccines against the efficiency of the activated vaccine Bartha-K61 and inactivated vaccine vPRV*. Pig experiments showed that the FJ-2012-VG* and FJ-2012-GEL* vaccines provided complete protection against an FJ-2102 strain challenge, whereas vPRV* provided protection with mild clinical signs, but the Bartha-K61 vaccine could not provide full protection, which may be due to a lower vaccine dose (5000 TCID_50_/dose). This indicated that the PRV FJ-2012 gE/gI deleted mutant inactivated vaccine, which is homologous with the virulent virus, provided better protection from the novel PRV, and is a promising candidate vaccine.

Until now, vaccines have been one of the best strategies for controlling PR [15]. Glycoproteins gE and gI correlate with PRV virulence, but they are not essential for viral replication. Furthermore, the deletion of these genes did not affect its immunogenicity, which provided an opportunity to develop vaccines that did not express gE/gI proteins responsible for PRV virulence [47,48,49]. In our report, the FJ-2012 strain was isolated from infected piglets vaccinated with Bartha-K61 and had been confirmed to belong to the same clade as the other PRV variants isolated after 2011 based on genomic sequence analysis. Moreover, the gB, gC, and gD genes of PRV FJ-2012 had significant mutations compared to the Bartha-K61 vaccine [38]. Considering that the gB, gC, and gD genes are the major neutralizing stimulation antigens of PRV, a vaccine constructed with gene deletion mutants from a homologous virulent virus could provide better protection; therefore, the PRV FJ-2012 strain was used as a parental strain to develop a PRV gene-deleted strain.

Taking safety and eradication into consideration, an inactive vaccine was generated instead of a live one in this study. Many studies revealed that a vaccine combined with an adjuvant had improved immunogenicity and immune response [50]. MONTANIDE^TM^ GEL02 is a range of ready-to-disperse innovative polymeric adjuvants: they have a slow release due to their polymer adsorption properties and they improve the recruitment of the innate immune system. GEL02 has been reported to enhance the vaccine immune response to porcine circovirus type 2 (PCV2), *Streptococcus agalactiae* (*S. agalactiae*), and *Erysipelothrix rhusiopathiae* (*E. rhusiopathiae*) [51,52,53]. ISA 206VG is a group of water-in-oil-in-water (W/O/W) adjuvants that induce short- and long-term protective immune responses because of their continuous aqueous phase emulsion structure. It is widely used as an adjuvant in foot-and-mouth disease (FMD) and poultry vaccines [54,55]. Here, both adjuvants were used to prepare the inactivated vaccines. Whether those prepared with adjuvant GEL02 or 206VG offer complete protection from a variant PRV challenge, the response time of the 206VG group of neutralizing antibodies and gB immune antibodies was faster. The reasons for the difference between the two adjuvants may be related to the structure of W/O/W, but this needs further study.

In commercial PRV vaccines, gE is commonly deleted, which is the reason that, based on serological tests, the PRV-gE ELISA kit was developed to distinguish vaccinated from wild-type PRV-infected animals [56], while gB antibody involvement in immunological levels was induced by immunization [57,58]. The piglets vaccinated with FJ-2012-VG* and FJ-2012-GEL* showed no clinical signs of PRV infection and generated gB-specific, but not gE-specific, ELISA antibodies. Therefore, FJ-2012-VG* and FJ-2012-GEL* are marker vaccines that could be used to control and eradicate PRV.

The VNA antibody level is an important indicator of vaccine efficacy and is related to protective efficacy [59]. Here, the FJ-2012-VG* and FJ-2012-GEL* vaccines developed higher titers of neutralizing antibodies against FJ-2012 compared to the Bartha-K61 vaccine at 42 dpc (*p* < 0.01). After the challenge, all piglets vaccinated with FJ-2012-VG* and FJ-2012-GEL* showed a transient increase in rectal temperature but no other clinical PR symptoms. All survived and were fully protected. However, the piglets in the Bartha-K61 group generated limited neutralizing antibody titers against FJ-2012, displayed 20% mortality after the FJ-2012 challenge, had a high rectal temperature, and showed severe clinical signs of depression, anorexia, and respiratory distress. In addition, after booster immunization, the vPRV* group also generated higher neutralizing antibody titers at 42 dpc compared with Bartha-K61 (*p* < 0.01). All piglets survived but had slight PR clinical signs. Therefore, FJ-2012-VG* and FJ-2012-GEL* provided better protection from novel PRVs compared to Bartha-K61, which is consistent with previous research that found that Bartha-K61 could not provide full protection against a PRV variant challenge [24,42,60].

Furthermore, the FJ-2012-VG* and FJ-2012-GEL* vaccines caused no visible gross pathological lesions in the organs of immunized piglets, but severe pathological lesions were observed in the Bartha-K61 and unimmunized challenged groups, and slight hemorrhages and necrosis were found in the lungs of the vPRV* group. Viral load is one of the important parameters used to evaluate vaccine efficacy [61]. The viral genomic copy numbers of tissue samples in the FJ-2012-VG* and FJ-2012-GEL* vaccine group decreased significantly (*p* < 0.01), and the unimmunized challenged group showed excessive copies in the brain, lung, and tonsils. This suggested that FJ-2012-VG* and FJ-2012-GEL* are effective vaccines, which was consistent with the findings previously reported [28].

The Bartha-K61 vaccine is widely used the world over and seems to be a long-standing tested vaccine [34]. In the field, incomplete protection may be influenced by the use of poor quality vaccines, improper storage, poor biosecurity, improper immunization schedules, or co-infections with other farm pathogens (such as PRRSV or CSF) [33]. In the present study, incomplete protection from the Bartha-K61 vaccine may have been due to a lower vaccine dose (5000 TCID_50_/dose). As previously reported, piglets vaccinated with a dose of 10^6.3^ TCID_50_ were protected against the PRV variant (XJ5) challenge, while no significant differences were observed between single and prime-boost vaccinated piglets [35]. This appearance implied that a lower vaccine dose may be the main reason for protection failure, but the detailed mechanism of incomplete protection remains unclear and needs further investigation.

## 5. Conclusions

The inactivated FJ-2012-VG* and FJ-2012-GEL* vaccines were effective and provided full protection against the FJ-2012 challenge which is highly homologous with a virulent virus. Moreover, the gE/gI-deleted vaccine could be differentiated from infected animals via ELISA in serological analysis. Thus, the vaccines in this study are promising as therapeutic strategies to control or eradicate the new PRV epidemic in China.

## Figures and Tables

**Figure 1 microorganisms-10-01880-f001:**
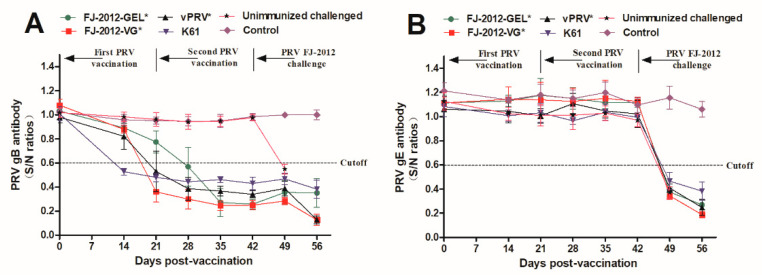
Development of anti-gB (**A**) and gE (**B**) antibodies with ELISA. (**A**) gB-specific antibody levels: a sample with S/N ratios ≤ 0.60 were classified as positive for gB antibodies. (**B**) gE-specific antibody levels: a sample with S/N ratios ≤ 0.60 were classified as positive for gE antibodies. The “*” mark in the group top right indicates an inactivated vaccine.

**Figure 2 microorganisms-10-01880-f002:**
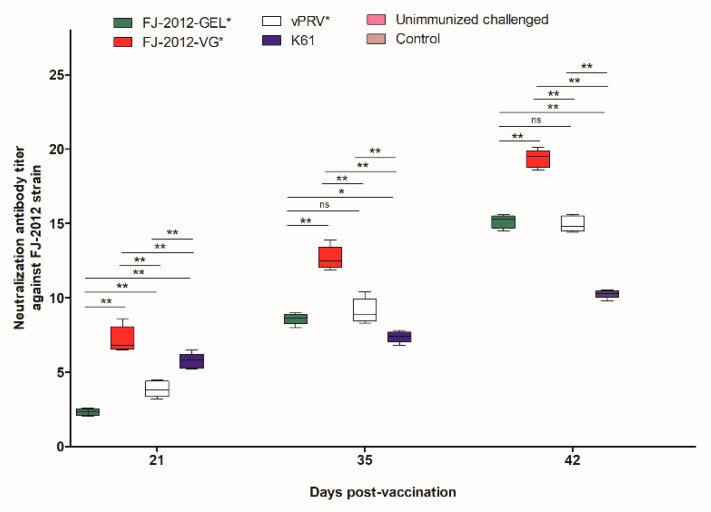
The neutralizing antibody titers of immunized piglets (*n* = 5) in a vaccination group. At 21, 35, and 42 days post-immunization, serum samples were collected and tested by SVNT. Statistical analysis was performed using GraphPad Prism version 6.0 for Windows (Dr. Harvey Motulsky, GraphPad Software, CA, USA). Pairwise comparisons of neutralizing antibody titers between different groups were performed by one-way ANOVA, followed by Tukey’s multiple comparison test. Significant differences (*p* < 0.01) are marked by **; (*p* < 0.05) by *; and no significant difference by ns.

**Figure 3 microorganisms-10-01880-f003:**
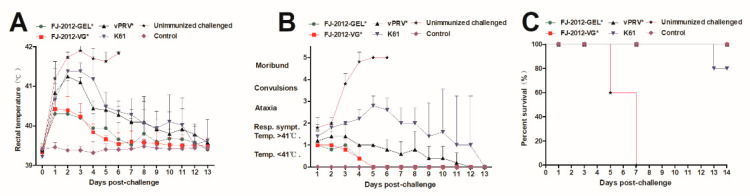
Protection efficacy of different vaccines against PRV FJ-2012 challenge in piglets. (**A**) Rectal temperature of piglets after challenging with FJ-2012. (**B**) Clinical scores with mean values and standard deviation. (**C**) Percentage survival curve of piglets vaccinated with a different vaccine. Survival is presented as a Kaplan–Meier plot (*n* = 5 per group). The “*” mark in the group top right indicates an inactivated vaccine.

**Figure 4 microorganisms-10-01880-f004:**
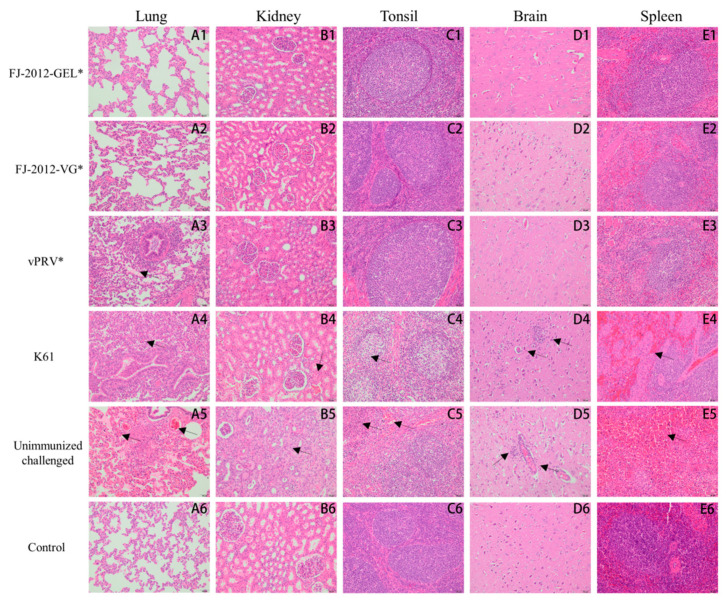
Histopathological findings of piglet tissues after PRV challenge. The letters A to E refer to the lung, kidney, tonsil, brain, and spleen, respectively. The numbers 1 to 6 represent the FJ-2012-GEL*, FJ-2012-VG*, vPRV*, K61, unimmunized challenged, and control groups, respectively. Hematoxylin and eosin staining (H.E.), Magnification, 200×. The “*” mark in the group top right indicates an inactivated vaccine.

**Figure 5 microorganisms-10-01880-f005:**
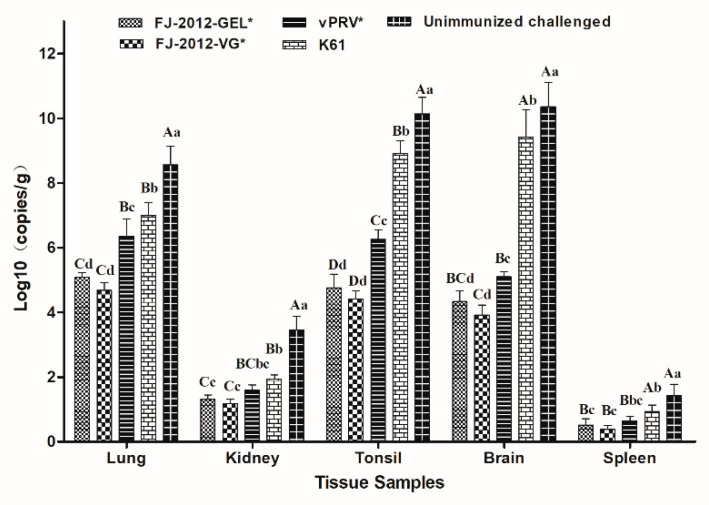
Viral load in different tissue samples from piglets after challenge. Viral genome numbers are presented as mean ± SD as measured by one-way repeated measurement analysis of variance (ANOVA) and least significant difference (LSD); the uppercase letters on the group bar indicate that the difference was extremely significant (*p* < 0.01); lowercase letters indicate a significant difference (*p* < 0.05). The “*” mark in the group top right indicates an inactivated vaccine.

**Table 1 microorganisms-10-01880-t001:** List of genetically modified vaccines based on emerging PRV in China.

Vaccine Type	Gene-Deleted Vaccines	Vaccine Strain	Phenotype	References
Inactivated	Single gene-deleted	AH02LA, HN1201 (licensed)	gE-deleted	[18,19]
	Double gene-deleted	ZJ01	gE/gI-deleted	[20]
Live-attenuated	Single gene-deleted	TJ	gE-deleted	[21]
	Double gene-deleted	JS-2012	gE/gI-deleted	[22]
	Double gene-deleted	JS-2012	gE/US2-deleted	[23]
	Double gene-deleted	XJ, AH02LA	gE/TK-deleted	[24,25]
	Triple-gene-deleted	ZJ01, MX, HN1201, HeN1, NY, and TJ	gE/gI/TK-deleted	[26,27,28,29,30,31]
	Four-gene-deleted (natural losses)	C (licensed)	gE/gI/US9/US2-deleted	[32]

**Table 2 microorganisms-10-01880-t002:** Schedule and timepoint design of immunization and challenge in different groups.

Groups	Immunization Time		Challenge Time
	0 Days	21 Days	42 Days
FJ-2012-GEL*	√	√	√
FJ-2012-VG*	√	√	√
vPRV*	√	√	√
K61	√	×	√
Unimmunized challenged	√	√	√
Control	×	×	×

Note: The “*” mark in the top right indicates an inactivated vaccine, and no mark indicates a live-attenuated vaccine; “√” indicates immunization or challenge; “×” indicates untreated.

**Table 3 microorganisms-10-01880-t003:** The incidence rate of tissue lesions in piglets after challenge with the PRV FJ-2012.

Groups	Lung	Kidney	Tonsil	Brain	Spleen
FJ-2012-GEL*	0 ^a^/5 ^b^	0/5	0/5	0/5	0/5
FJ-2012-VG*	0/5	0/5	0/5	0/5	0/5
vPRV*	2/5	0/5	0/5	0/5	0/5
K61	3/5	0/5	2/5	1/5	0/5
Unimmunized challenged	5/5	3/5	3/5	5/5	2/5
Control	0/5	0/5	0/5	0/5	0/5

^a^ Number of piglets positive for tissue lesions. ^b^ Number of piglets tested. The “*” mark in the group top right indicates an inactivated vaccine.

**Table 4 microorganisms-10-01880-t004:** Microscopic lesion scores of tissue in piglets after challenge with the PRV FJ-2012.

Groups	MLS ± SD ^§^				
	Lung ^†^	Kidney ^‡^	Tonsil ^†^	Brain ^†^	Spleen ^†^
FJ-2012-GEL*	0.00 ± 0.00 ^Bc‖^	0.00 ± 0.00 ^Bc^	0.00 ± 0.00 ^Cc^	0.00 ± 0.00 ^Bc^	0.00 ± 0.00 ^Bc^
FJ-2012-VG*	0.00 ± 0.00 ^Bc^	0.00 ± 0.00 ^Bc^	0.00 ± 0.00 ^Cc^	0.00 ± 0.00 ^Bc^	0.00 ± 0.00 ^Bc^
vPRV*	2.00 ± 1.00 ^BCb^	0.00 ± 0.00 ^Bc^	0.00 ± 0.00 ^Cc^	0.00 ± 0.00 ^Bc^	0.00 ± 0.00 ^Bc^
K61	3.67 ± 1.15 ^ACb^	1.33 ± 0.58 ^Ab^	2.33 ± 0.58 ^Bb^	1.67 ± 1.16 ^Bb^	2.33 ± 1.53 ^ABb^
Unimmunized challenged	5.67 ± 0.58 ^Aa^	2.33 ± 0.58 ^Aa^	5.00 ± 1.00 ^Aa^	4.67 ± 1.16 ^Aa^	4.33 ± 0.58 ^Aa^
Control	0.00 ± 0.00 ^Bc^	0.00 ± 0.00 ^Bc^	0.00 ± 0.00 ^Cc^	0.00 ± 0.00 ^Bc^	0.00 ± 0.00 ^Bc^

^§^ Mean lesion score (MLS) ± standard deviations (SD), measured by one-way repeated measurement analysis of variance (ANOVA) and least significant difference (LSD). ^†^ Score range from 0 (normal) to 6 (severe). ^‡^ Score range from 0 (normal) to 3 (severe). ^‖^ The uppercase letters in the columns indicate that the difference was extremely significant (*p* < 0.01); lowercase letters indicate a significant difference of (*p* < 0.05). The “*” mark in the group top right indicates an inactivated vaccine.

## Data Availability

All data are reported in this article.

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
