# Peer review of "The Immune Efficacy of Inactivated Pseudorabies Vaccine Prepared from FJ-2012ΔgE/gI Strain"

_microorganisms, 2022, doi:10.3390/microorganisms10101880_

Round 1
Reviewer 1 Report
General comments:
The Bartha K61 vaccine remains a golden standard vaccine in various AD eradication programs to this day. Since the recent emergence of new virulent pseudorabies virus (PRV) strains in vaccinated pig farms in China, however, insufficient efficacy and protection of this particular vaccine has been claimed. While crucial information is missing on how Chinese vaccination programs were performed as these are not mandatory and information about the quality of locally licensed and produced Bartha-K61 vaccines is not always readily available, experimental assays suggesting incomplete protection provided by Bartha K61 against the novel Chinese PRV strains were often performed in non-target species that are difficult to extrapolate to pigs. In this manuscript the authors present results from an experimental study showing that two differently adjuvanted forms of the inactivated pseudorabies vaccine (FJ-2012ΔgE/gI- strain) protect piglets from infection with the newly emerged Chinese PRV strains.
I have a number of points to address that lack necessary diligence and scientific thoroughness and hence, require revision of almost all parts of the manuscript.
Introduction
- In large parts, the introduction does not constitute meaningful information and is completely ignorant of international literature on RRV vaccine developments. One should be able to expect mentioning of which types of PRV vaccines are available with a corresponding reference to the relevant literature. This should include "novel" Chinese PRV vaccine developments published recently as well.
- The authors fail to demonstrate the need on vaccine research. The alleged failure of the Bartha K61 vaccine is definitely not a convincing argument, as recent papers show that this vaccine strain is protective even against the newly emeerged Chines PRV variants. So what was the rational in developing new vaccine constructs?
- Lines 36-39: Please, delete these sentences (also in the Discussion). Pseudorabies is definitely not a zoonosis and the cited papers including the results are no proof!
- Line 44-46: What is meant with antigenicity here?
- Lines 46-48: This contradictes published reports from European countries, USA and Canada, which succesfully used the Bartha strain for PRV elimination. Even recent literature using contradicts this hypothesis (not cited at all). So, please, consider for revision and disucssion.
- I do not agree that the authors have investigated the safety of the vaccine according to international standards.
Materials and Methods
- Lines 64-66: Please, explain why no SPF animals were used. Can the authors exclude any impact from diseases other than those they have tested for, e.g. PRV, CSF and PRRS, on the results obtained?
- Lines 70: Please provide details on the adjuvants used.
- Lines 71-73: Please indicate the manufacturer of the commercial Bartha K61 vaccine and the inactivated PRV-Ea strain used, as is the case for diagnostic test kits or reagents.
- Line 86: The unimmunized group seems to be a placebo group?
- Lines 88 -90, Table 1: Why is there a difference in the vaccination schedule, with the two experimental and the PRV Ea vaccines given 2ml at days 0 and 21 versus Bartha K61 given 1 dose only at day 0? Do the schedules for the commercial PRV vaccines reflect manufacturers instructions? Are the antigen contents of the inactivated vaccines comparable? What was the vaccine virus titre of the attenuated Bartha-K61 vaccine and was the titre verfied before administration? Please, clarify and discuss in the light of the results obtained. Also, it is not clear for the reader here that the Bartha-K61 vaccines is an attenuated one. Besides, this is redundant information in text and table.
- Lines 94-97: How were clinical signs objectively assessed. Was a clinical scoring scheme applied? Otherwise this is very subjective. Also, as this was a severe challenge, from an animal welfare point of view what was the humane endpoint? Was there any defined at all?
- Lines 99-105: Why is there a difference in sampling scheme for the detction of binding and virus neutralizing antibodies. Please, clarify. For reasons of comparability, it would be good if the results of both serological tests were available for all sampling time points.
- Lines 112-137: What was the rational for histological examination of and virus detection via PCR in challenged animals (objectives?). Please, clarify.
- Lines 130-136: Please, describe the method of establishing genome copies in more detail. How to get from the amplification kinetic curve to genome copies per gramm.
- Lines 140-142: Please, include abbreviation for one way repeated measurement analysis of variance (ANOVA).
Results
- Lines 151-160: Please, rephrase the paragraph. You are presenting S/N ratios which is equivalent to gB and gE specific binding antibodies. For the reader it hard to follow an increase in antibodies as depicted in Figure 1. From what can be obtained from figure 1A the slight decrease in antibody response on day 49 and the subsequent gradual increase is not real but within the normal range of variation. Please, change.
- Line 171: Please, use correct technical terms: Virus neutralising antibodies (VNA)
- Lines 174-181: Are the differences in VNAs between the vaccinated groups statistically validated?
- Figure 2: Instead of three separate graphs, use one graph that shows the direct comparison in the development of virus neutralising antibodies between the different experimental groups. Box plots are better suited for this purpose. What is represented by the horizontal lines?
- Lines 214-245: Given the fact that these pigs used were no SPF animals (but originated from a family farm) how do you know that all these pathological gross and histological lesions are due to PRV infections? Please, discuss.
Discussion
- The Discussion needs imporvement. Not all of the results obtained are properly interpreted, nor are the findings compared and discussed to those of similar studies. The authors fail to argue how the answers fit in with existing knowledge on the topic. Here, the main focus should be on efficacy and the advantages the two adjuvanted PRV vaccine constructs offer in comparison to others if any. Also, limitations of the study are not addressed. Please avoid repition of results.
- Lines 324-330: Please refer to the comment on the Bartha vaccine virus strain above (Introduction) and discuss factually. Also, can the different vaccination schedule, antigen content etc. have an influence on the results obtained with the commercial PRV vaccines (see comments on Material and Methods)?
General notes
- The labelling of the different experimental groups is to small. Please, use larger characters.
- The figure captures need improvement. Figure captions should be standalone, i.e., descriptive enough to be understood without having to refer to the main text.
- Often appropriate references (in particular primary literature) are missing or references are even inappropriate.
- Extensive editing of English language and style required by a native speaker or editing service.
Reviewer 2 Report
The paper will need proofreading which includes checking for correct grammar, correct spelling and comprehension.
The flow and interpretation of data will then be clarified.
Author Response
Thank you for your suggestion, and we had checked the grammar,spelling and comprehension full paper. And the flow and interpretation of data had been clarified.
Reviewer 3 Report
In my opinion, the English format throughout the entire manuscript must be carefully checked and dramatically improved before further considering the present paper for publication.
Author Response
Thank you for your suggestion, we had checked and improved the English format in the manuscript.
Round 2
Reviewer 1 Report
One must acknowledge the authors' efforts to improve the manuscript. Substantial changes have been made that meet scientific requirements.
I have only a few minor points for further consideration.
The following fundamental publications should still be mentioned in the introduction or discussion:
- Freuling CM, Müller TF, Mettenleiter TC. Vaccines against pseudorabies virus (PrV). Vet Microbiol. 2017 Jul;206:3-9.
- Delva JL, Nauwynck HJ, Mettenleiter TC, Favoreel HW.The Attenuated Pseudorabies Virus Vaccine Strain Bartha K61: A Brief Review on the Knowledge Gathered During 60 Years of Research.Pathogens. 2020 Oct 27;9(11):897.
Please, replace the abbreviation PR with PRV throughout the manuscript.
Please, mention the TCID50 of the comemrcial Bartha strain used for comparison in Materials and Methods.
Author Response
Dear reviewer,
Thank you for your comments, and we had performed all the suggestions. The finished response as following:
Comment 1.
The following fundamental publications should still be mentioned in the introduction or discussion:
- Freuling CM, Müller TF, Mettenleiter TC. Vaccines against pseudorabies virus (PrV). Vet Microbiol. 2017 Jul;206:3-9.
- Delva JL, Nauwynck HJ, Mettenleiter TC, Favoreel HW.The Attenuated Pseudorabies Virus Vaccine Strain Bartha K61: A Brief Review on the Knowledge Gathered During 60 Years of Research.Pathogens. 2020 Oct 27;9(11):897.
Response 1.
We had mentioned the references in the introduction or discussion. Line 55, 366 and line 62, 357, and 427.
Comment 2.
Please, replace the abbreviation PR with PRV throughout the manuscript.
Response 2.
We have corrected the PR in full text. Besides, PR: pseudorabies, PRV: Pseudorabies virus, Pseudorabies virus (PRV) is the pathogen that causes pseudorabies (PR).
Comment 3.
Please, mention the TCID50 of the comemrcial Bartha strain used for comparison in Materials and Methods.
Response 3.
We had mentioned the TCID50 of the comemrcial Bartha strain. Lines 87.
Reviewer 2 Report
The revised version is more reader-friendly. It is an interesting study and I only have a few suggestions.
Introduction, line 47: Change "engineering" to "engineered".
Introduction, line 67: Please change "bad" to "had".
Table 1, notes: Did you mean attenuated and not "activated" as I never heard of activated vaccines.
Materials and methods, 2.7, line 156: Why did you add an elongation step? In which step is the flourescence for Ct-value captured?
Although Figure 5 indicate the changes in or comparison of virus recovered. Please confirm calculation method for virus load.
Discussion, line 333: Please write a reason such as did not show protection after "Bartha-K61 vaccine did not."
Discussion, line 369-370: It will not be possible to "eradicate PRV via the gE/gB ELISA kit." This will be used as a differential test to determine vaccinated vs unvaccinated animals.
Author Response
Dear reviewer,
Thank you for your comments, and we had performed all the suggestions. The finished response as following:
Comment 1.
Introduction, line 47: Change "engineering" to "engineered".
Response 1.
We have changed.
Comment 2.
Introduction, line 67: Please change "bad" to "had".
Response 2.
We have changed.
Comment 3.
Table 1, notes: Did you mean attenuated and not "activated" as I never heard of activated vaccines.
Response 3.
We have changed. That’s mean attenuated vaccine.
Comment 4.
Materials and methods, 2.7, line 156: Why did you add an elongation step? In which step is the flourescence for Ct-value captured?
Response 4.
I’m sorry, the elongation step is not need. We have changed. And the annealing at 60 ℃ is the flourescence for Ct-value captured.
Comment 5.
Although Figure 5 indicate the changes in or comparison of virus recovered. Please confirm calculation method for virus load.
Response 5.
The viral load in the tissue samples was estimated following the references:
[1] Gu, Z.; Dong, J.; Wang, J.; Hou, C.; Sun, H.; Yang, W.; Bai, J.; Jiang, P. A novel inactivated gE/gI deleted pseudorabies virus (PRV) vaccine completely protects pigs from an emerged variant PRV challenge. Virus research 2015, 195, 57-63, doi:10.1016/j.virusres.2014.09.003.
[2] Zhao, Y.; Wang, L.Q.; Zheng, H.H.; Yang, Y.R.; Liu, F.; Zheng, L.L.; Jin, Y.; Chen, H.Y. Construction and immunogenicity of a gE/gI/TK-deleted PRV based on porcine pseudorabies virus variant. Molecular and cellular probes 2020, 53, 101605, doi:10.1016/j.mcp.2020.101605.
Comment 6.
Discussion, line 333: Please write a reason such as did not show protection after "Bartha-K61 vaccine did not."
Response 6.
We have supplemented. Line 363.
Comment 7.
Discussion, line 369-370: It will not be possible to "eradicate PRV via the gE/gB ELISA kit." This will be used as a differential test to determine vaccinated vs unvaccinated animals.
Response 7.
This mean that FJ-2012-VG* and FJ-2012-GEL* are marker vaccines that could be used to control and eradicate PRV using the strategy of vaccination-DIVA testing, i.e.the serological differentiation of vaccinated from field-virus infected pigs by the use of marker vaccines and respective companion diagnostic tests.
Reviewer 3 Report
Although improved, I consider that the paper by Chen et al. is rather inaccurate and still unsuitable for publication in the present format.
Comments
TITLE – I suggest to deeply modify the title, which sounds as a too sharp and definitive conclusion in the present format. Moreover, I suggest to avoid abbreviation (PRV) at this point.
ABSTRACT – In my opinion, it should be entirely re-written. In the present formant, it is too confuse and inaccurate. Examples:
Line 25 - “In contrast, all of the piglets in the piglets in unimmunized group were all sacrificed”. The meaning of this sentence is somewhat obscure. Actually, all piglets under study were finally sacrificed. Thus, what do Authors mean for “sacrifice”?
Line 26 - “vPRV*” At this point, it is impossible to understand the meaning of asterisk.
Line 27 – “clinical symptoms” should be corrected as “clinical signs” throughout the entire manuscript.
KEYWORDS
Inactive? Do Author mean “inactivated”?
Effective? Maybe, efficacy? Vaccine efficacy?
INTRODUCTION
Lines 52-54 – almost incomprehensible. Moreover, suitable references should be added.
Lines 58-60 – almost incomprehensible.
Lines 66-81 – such data might be better explained in a Table.
Lines 81-87 – quite incomprehensible.
MATERIALS AND METHODS
Lines 111-117 – further data should be provided about laboratory tests, which were performed to rule out CSFV, PRV and PRRSV infections.
Line 122 – “Bartha strain” is wrongly reported in several lines of the manuscript.
TABLE 1 – incomprehensible.
Lines 138-147 – not easy to read and to understand in the present format.
Line 149 – “initial” immunization? Please, be more precise.
Lines 149-156 – Who carried out the clinical observations/scores? One or more operators. Were they blind to the experimental conditions? Please, add more details.
Line 166 – “serum was collected”?
Line 176 – please, add more details about euthanasia protocol.
Line 178 – why sampling was restricted to those 5 tissues/organs? Did Author refer to previous papers? Why they did not consider further tissues, such as nasal mucosa, trigeminal ganglion etc?
RESULTS
Line 227 – “spirits”?
Line 242 – please, delete “as expected”.
Lines 275-281 – these sentences must be deleted, as they deal with “materials and methods”.
Line 289 – “A clinical score only began at 2 and 3 dpc”. Please, better explain this sentence.
The terms “fortunately” and “unfortunately” sound inadequate.
The way clinical signs have been scored and calculated is not clear to me. Please, provide further details.
Lines 317- 355 – pathological findings are poorly described. Moreover, some pictures are unsuitable to show such microscopic findings (too low magnification). Authors state that lesions were “milder” in some groups, or “severe”, or “no significant”. However, it is not clear who carried out such evaluation (one or more operators? Blind to the experimental conditions?). Did Authors follow a scoring protocol for lesions? Some pathological findings could result from euthanasia; please, comment these points.
DISCUSSION
Line 452 – What do Authors mean for “complete” protection?
Line 470 – Attenuated vaccines have been/are widely used, as they prove to be safe and more effective, when compared with inactivated vaccines. Please, discuss this point.
Line 475 – Streptococcus? Which one?
Line 500 – 80% mortality? Please, check this data.
Author Response
Dear reviewer,
Thank you for your comments, and we had performed all the suggestions. The finished response as following:
Comment 1.
TITLE – I suggest to deeply modify the title, which sounds as a too sharp and definitive conclusion in the present format. Moreover, I suggest to avoid abbreviation (PRV) at this point.
Response 1.
We have changed the manuscript title, follow:
The immune efficacy of inactivated pseudorabies vaccine prepared from FJ-2012ΔgE/gI strain.
Comment 2.
ABSTRACT – In my opinion, it should be entirely re-written. In the present formant, it is too confuse and inaccurate. Examples:
Line 25 - “In contrast, all of the piglets in the piglets in unimmunized group were all sacrificed”. The meaning of this sentence is somewhat obscure. Actually, all piglets under study were finally sacrificed. Thus, what do Authors mean for “sacrifice”?
Line 26 - “vPRV*” At this point, it is impossible to understand the meaning of asterisk.
Line 27 – “clinical symptoms” should be corrected as “clinical signs” throughout the entire manuscript.
Response 2.
We have re-written the abstract.
The asterisk denotes inactivation.
And the “clinical symptoms” have corrected as “clinical signs” throughout the entire manuscript.
Comment 3.
KEYWORDS
Inactive? Do Author mean “inactivated”?
Effective? Maybe, efficacy? Vaccine efficacy?
Response 3.
We have corrected the keywords. That’s mean “inactivated” and Vaccine efficacy
INTRODUCTION
Comment 4.
Lines 52-54 – almost incomprehensible. Moreover, suitable references should be added.
Response 4.
This mean that PRV vaccines types, including inactivated, live-attenuated, subunit, and live virus-vectored vaccine. But we re-written this part, and the sentence deleted.
Comment 5.
Lines 58-60 – almost incomprehensible.
Response 5.
That is mean: PR has been largely controlled by the wide use of inactivated and live-attenuated vaccines, and some farms eradicated the PR via concept of DIVA, i.e. the serological differentiation of vaccinated from field-virus infected pigs by the use of marker vaccines and respective companion diagnostic tests.
Comment 6.
Lines 66-81 – such data might be better explained in a Table.
Response 6.
We have summarized in Table 2.
Comment 7.
Lines 81-87 – quite incomprehensible.
Response 7.
That is mean: PRV has been developed into a powerful vector system for foreign proteins to simultaneously vaccinate against several animal infectious diseases. And recombinant expressing gD or gC/gD-based proteins vaccines.
MATERIALS AND METHODS
Comment 8.
Lines 111-117 – further data should be provided about laboratory tests, which were performed to rule out CSFV, PRV and PRRSV infections.
Response 8.
We have provided the results of test. See the attachment.
Comment 9.
Line 122 – “Bartha strain” is wrongly reported in several lines of the manuscript.
Response 9.
We have corrected.
Comment 10.
TABLE 1 – incomprehensible.
Response 10.
We have corrected the TABLE 1. This is mean that the schedule and timepoint design of the vaccination, challenged in different groups.
Comment 11.
Lines 138-147 – not easy to read and to understand in the present format.
Response 11.
We have re-written this part in Materials and Methods 2.3. Line 111-120
Comment 12.
Line 149 – “initial” immunization? Please, be more precise.
Response 12.
We have correct “initial”, and use “first” instead.
Comment 13.
Lines 149-156 – Who carried out the clinical observations/scores? One or more operators. Were they blind to the experimental conditions? Please, add more details.
Response 13.
There are two operators to carried out the clinical observations/scores,
After challenged, survival rate and rectal temperatures were measured once daily (morning), and clinical signs of each piglet were observed and scored twice (morning/evening) daily according to the previous report [39]: clinical scores were as-sessed as (1) elevated temperature above 40 °C and below 41 °C; (2) fever above 41 °C combined with respiratory distress; (3) ataxia; (4) convulsions; and (5) moribund or dead. The higher of the two scores each day was assigned as the individual daily score.
Comment 14.
Line 166 – “serum was collected”?
Response 14.
We have corrected the sentence. “Serum samples was taken from…”. Line 128.
Comment 15.
Line 176 – please, add more details about euthanasia protocol.
Response 15.
We have add the details about euthanasia protocol. At 14 days post-challenge (dpc), all surviving pigs were euthanized by injections of tiletamine hydrochloride and zolazepam hydrochloride (Virbac, Carros, France). A complete necropsy was performed within 2 h of death, and tissue samples required for histological examination. Line 137-139.
Comment 16.
Line 178 – why sampling was restricted to those 5 tissues/organs? Did Author refer to previous papers? Why they did not consider further tissues, such as nasal mucosa, trigeminal ganglion etc?
Response 16.
We have refer to the previous paper follow:
[1] Gu, Z.; Dong, J.; Wang, J.; Hou, C.; Sun, H.; Yang, W.; Bai, J.; Jiang, P. A novel inactivated gE/gI deleted pseudorabies virus (PRV) vaccine completely protects pigs from an emerged variant PRV challenge. Virus research 2015, 195, 57-63, doi:10.1016/j.virusres.2014.09.003.
[1] Li, W., Zhuang, D., Li, H., Zhao, M., Zhu, E., Xie, B., Chen, J., & Zhao, M. (2021). Recombinant pseudorabies virus with gI/gE deletion generated by overlapping polymerase chain reaction and homologous recombination technology induces protection against the PRV variant PRV-GD2013. BMC veterinary research, 17(1), 164. https://doi.org/10.1186/s12917-021-02861-6
RESULTS
Comment 17.
Line 227 – “spirits”?
Response 17.
We have deleted the “spirits”, it may be a repeat statement with “displayed no clinical signs”.
Comment 18.
Line 242 – please, delete “as expected”.
Response 18.
We have deleted.
Comment 19.
Lines 275-281 – these sentences must be deleted, as they deal with “materials and methods”.
Response 19.
We have deleted.
Comment 20.
Line 289 – “A clinical score only began at 2 and 3 dpc”. Please, better explain this sentence.
Response 20.
The clinical score was only raised at 2 and 3 dpc.
Comment 21.
The terms “fortunately” and “unfortunately” sound inadequate.
Response 21.
We have corrected throughout the entire manuscript.
Comment 22.
The way clinical signs have been scored and calculated is not clear to me. Please, provide further details.
Response 22.
There are two operators to carried out the clinical observations/scores,
After challenged, survival rate and rectal temperatures were measured once daily (morning), and clinical signs of each piglet were observed and scored twice (morning/evening) daily according to the previous report [39]: clinical scores were assessed as (1) elevated temperature above 40 °C and below 41 °C; (2) fever above 41 °C combined with respiratory distress; (3) ataxia; (4) convulsions; and (5) moribund or dead. The higher of the two scores each day was assigned as the individual daily score.
Comment 23.
Lines 317- 355 – pathological findings are poorly described. Moreover, some pictures are unsuitable to show such microscopic findings (too low magnification). Authors state that lesions were “milder” in some groups, or “severe”, or “no significant”. However, it is not clear who carried out such evaluation (one or more operators? Blind to the experimental conditions?). Did Authors follow a scoring protocol for lesions? Some pathological findings could result from euthanasia; please, comment these points.
Response 23.
We have supplemented the scoring protocol for lesions in the method (part 2.6, Line 136-161), and the result was summarized in Table 3 and Table 4.
In addition, we have re-written the part of 3.4 Necropsy and histological analysis.
Consider to the pathological findings could result from the euthanasia, we used the RT-PCR to future confirm the pathological findings result from PRV challenged. The results of pathological findings and RT-PCR are consistent.
Thus, we cannot completely exclude the influence caused by euthanasia, but we have chose the drug that had the least impact on the pathology, minimized the influence of euthanasia.
The microscopic finding is magnification at 200X, and the LaTeX format (Unable to fit large images in text) resulting in smaller field of view. So, for the magnification in this figure, the reader is referred to the web version of this article.
DISCUSSION
Comment 24.
Line 452 – What do Authors mean for “complete” protection?
Response 24.
This mean that “full protection” or “sufficient protection”, we have corrected.
Comment 25.
Line 470 – Attenuated vaccines have been/are widely used, as they prove to be safe and more effective, when compared with inactivated vaccines. Please, discuss this point.
Response 25.
The important one is the live-attenuated vaccine safety to pig populations should be confirmed via lengthy field testing. Taken the safety of the vaccine candidate into consideration, the vaccine strain used in this study was an inactivated instead of a live vaccine.
Comment 26.
Line 475 – Streptococcus? Which one?
Response 26.
Streptococcus agalactiae (S. agalactiae)
Comment 27.
Line 500 – 80% mortality? Please, check this data.
Response 27.
I’m sorry, that is 20% mortality. We have corrected.